# MaskTab: Masked Tabular Data Modeling for Learning with Missing Features

## Abstract

Tabular machine learning has garnered increasing attention due to its practical value. Unlike the complete and standardized data often assumed in academia, tabular data primarily originates from industrial contexts and usually faces the issue of incomplete data samples, *i.e.*, some features of a sample may be unpredictably missing. In this work, we introduce MaskTab, a masked tabular data modeling framework designed to facilitate model learning despite missing features. Instead of pursuing to accurately restore missing features like existing imputation methods, we jointly approach missing feature modeling and downstream tasks (*e.g.*, classification) with a unified objective. Concretely, we propose to randomly drop out some solid features during training, equipped with a missing-related masked attention mechanism, to help the model rely more on trustworthy features when making decisions. Experiments on the very recent industry-grade benchmark, TabReD, suggest that our method surpasses the second DNN-based competitor by a clear margin, demonstrating its effectiveness and robustness in real-world scenarios. We will release the code and the model to facilitate reproduction.

## 1 Introduction

Machine learning (ML) models applied to tabular data have significant industrial applications, such as credit assessment (Aziz et al., 2022) and medical diagnosis (Hassan et al., 2020). Recent advancements in deep tabular models have demonstrated promising effectiveness in various academic scenarios. However, tabular data primarily originates from industrial contexts and usually faces the issue of incomplete data samples, which requires further investigation for modeling with missing features under industrial scenarios.

To address the issue of missing data, imputation methods (Du et al., 2024; Ma & Zhang, 2021; Jinsung Yoon & van der Schaar, 2019) have been developed to estimate missing values using observed data. However, these methods often prioritize accurate restoration of missing values and neglect the significance of imputation for downstream tasks, which can be suboptimal for prediction models, as indicated by recent studies (Daniel Jarrett & van der Schaar, 2022). Moreover, the assessment of these imputation methods typically involves synthetic data with simulated missing values, which may not adequately reflect the challenges posed by real-world data absence.

Recent advancements in deep tabular models have focused on the development of sophisticated network architectures, such as enhanced multilayer perceptrons (MLPs) and transformers (Chen et al., 2022; Gorishniy et al., 2021). Additionally, improvements in retrieval based methods have emerged, for instance, TabR (Gorishniy et al., 2024) has shown superior performance compared to XGBoost (Chen & Guestrin, 2016) on GBDT-friendly benchmarks (Grinsztajn et al., 2022). However, these approaches have predominantly been assessed on academic datasets characterized by low feature dimensionality and minimal missing values, which do not reflect the complexities of real-world industrial applications. Newly established industry-grade tabular benchmarks, TabReD (Rubachev et al., 2024), indicate that these advanced methods often underperform relative to more straightforward tree-based models when applied to industrial data. Notably, XGBoost continues to excel, particularly on the HomeCredit Default dataset, which characterized by significant missingness and dimensionality, greatly surpassing the performance of deep tabular models. This highlights the need to enhance the capabilities of deep tabular models in modeling missing features.

Another promising development in deep tabular models involves leveraging pre-training techniques, such as contrastive learning (Bahri et al., 2022; Somepalli et al., 2022), masked modeling (Ye et al., 2024), and technologies associated with large language models (Yan et al., 2024; Borisov et al., 2023). The pre-training of models to impart knowledge demonstrates significant potential in situations marked by limited samples and data scarcity.

In this work, we introduce MaskTab, a masked tabular data modeling framework designed to facilitate model learning despite missing features. Instead of pursuing to accurately restore missing features like existing imputation methods, we jointly approach missing feature modeling and downstream tasks (e.g., classification) with a unified objective. Concretely, we propose to randomly discarding essential features during the training process, which act as a simulation process to generate additional missing features in samples that are truly incomplete. Further, we introduce a joint learning framework, that integrates the reconstruction of simulated missing features with the optimization of downstream tasks, to learn a more robust embedding representation for truly missing values. We represent both true missing and simulated missing by initializing a parameter-shared, learnable mask embedding, to enhance the practical utility of the missing representation. Moreover, considering the joint learning framework, we emphasize the significance of masking strategies to minimize significant shifts in data distribution, as such shifts can negatively impact downstream task performance. We investigate the criteria for effective masking strategies and provide valuable insights applicable to diverse scenarios involving feature missingness. Additionally, we implement a missing-related masked attention mechanism, that integrates prior information of feature missingness into the attention computation, which enables the model to prioritize trustworthy features when making decisions for each sample. Experiments on the very recent industry-grade benchmark, TabReD, show the effectiveness of MaskTab in real-world industrial scenarios.

Our contributions are as follow:

- We propose MaskTab, a masked tabular data modeling framework to tackle the feature missing problem in industrial scenarios, experiments on TabReD demonstrate that our method surpasses the second DNN-based competitor by a clear margin.

- Ablation studies reveal that both the joint learning framework and the embedding representation specifically designed to handle genuinely missing values are effective. Additionally, the missing-related masked attention mechanism proves beneficial. Our findings provide significant insights applicable to a wide range of real-world scenarios.

- MaskTab employs an end-to-end training approach that eliminates the need for imputing missing features, which is both straightforward and effective.

## 2 RELATED WORK

**Masked Modeling.** Masked training was initially introduced as a pre-training strategy for language models. This technique occludes specific words in a sentence, requiring the model to predict the masked tokens based on the visible context. It has led to significant advancements in both natural language processing (NLP) and computer vision (CV), as exemplified by models like BERT (Devlin et al., 2018) and BEit (Bao et al., 2022). Recently, masked training has gained impact as a restoration pre-training task in the domain of tabular data. XTab (Zhu et al., 2023) and CM2 (Ye et al., 2024) employ masked learning on cross-table data to develop tabular foundation models. ReMask (Du et al., 2024) implementes missing feature imputation through optimizing the autoencoder by reconstructing a randomly re-masked features. TabMT (Gulati & Roysdon, 2023) designs advanced masking techniques to generate synthetic tabular data.

**Deep Tabular Models.** With advancements in deep neural networks, deep tabular models have transitioned from multilayer perceptrons (MLPs) to more sophisticated architectures, including ResNet (Gorishniy et al., 2021), SNN (Klambauer et al., 2017), DANets (Chen et al., 2022), and DCNv2 (Wang et al., 2020). These approaches typically operate directly on raw features. Transformer-based methodologies have garnered attention recently, which treat individual raw features as tokens and convert feature values into high-dimensional vectors using either lookup tables or linear mappings. Notable approaches including TransTab (Wang & Sun, 2022), AutoInt (Song et al., 2019), T2G-Former (Yan et al., 2023) and FT-Transformer (Gorishniy et al., 2021). These methods effectively utilize the attention mechanism to enhance feature interactions. Furthermore,

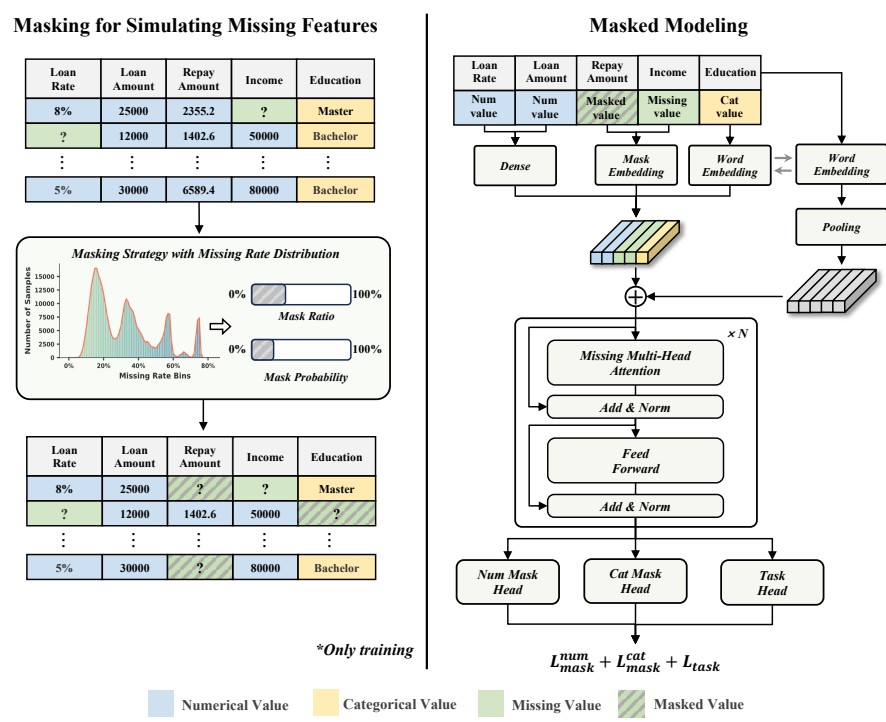

Figure 1: Overall framework of `MaskTab`. The left part delineates the masking for Simulating Missing Features, where the masking strategy is carefully designed with missing rate distribution of dataset. The right part provides an overview of masked modeling, including various data types feature embedding, transformer backbone and joint learning approach.

several studies have examined improved embedding representations for various data types. For instance, prior art (Gorishniy et al., 2022) enhances numerical representations, while CM2 (Ye et al., 2024) leverages word embeddings from a pre-trained BERT model to extract features from both categorical and textual data. Additionally, TP-BERT (Yan et al., 2024) introduces Relative Magnitude Tokens (RMT), enabling the language model to recognize relative dense value magnitudes within the language space. In addition, there are neighborhood-based methods, which involve extracting features from test samples and performing feature retrieval across the entire training dataset to identify the nearest neighbors. Regression or classification is then conducted using these neighbors. Notable examples of such methods include DNNR (Nader et al., 2022) and TabR (Gorishniy et al., 2024).

## 3 MASKTAB

We present `MaskTab`, as depicted in Fig. 1, we detail the method in this section. First, we outline the task formalization and the masked modeling for missing features, next, we present the feature embedding module, followed by the design of the tabular transformer in feature missing scenarios, finally, we detail the joint training approach for modeling missing features and downstream tasks.

### 3.1 TASK FORMALIZATION

Let us consider a dataset represented as $D = (\mathbf{x}_i, y_i)_{i=1}^n$, where $n$ indicates the total number of samples. Each sample $\mathbf{x}_i$ is composed of three distinct components: $\mathbf{x}_i = \{\mathbf{x}_i^{\text{cat}}, \mathbf{x}_i^{\text{num}}, \mathbf{x}_i^{\text{miss}}\}$. Specifically, $\mathbf{x}_i^{\text{cat}} = \{x_i^1, x_i^2, \ldots, x_i^a\}$ corresponds to $a$ categorical or textual features, while $\mathbf{x}_i^{\text{num}} \in \mathbb{R}^b$ denotes $b$ numerical features. Furthermore, $\mathbf{x}_i^{\text{miss}} = \{nan_i^1, nan_i^2, \ldots, nan_i^k\}$ accounts for $k$ missing features within the data. All samples within the dataset share equivalent tabular headers, which collectively constitute the feature names denoted by $\mathbf{C}_i = \{c^1, c^2, \ldots, c^{a+b+k}\}$.

Our objective is to enhance the accuracy of predicting $y_i$ in scenarios involving missing features. To simplify the notation, we omit the sample index $i$ in the descriptions of the methods that follow.

## 3.2 Masked modeling for Missing features

We employ a masked tabular data modeling framework to enhance the model's effectiveness in scenarios with feature missingness. Given a sample with truly missing features $\mathbf{x} = \{\mathbf{x}^{cat}, \mathbf{x}^{num}, \mathbf{x}^{miss}\}$, we employ a defined probability, referred to as $mask\_prob$ (e.g., 15%), to determine whether to simulate an increased level of feature missingness for it. For the samples that require simulation, we drop features—excluding those that are truly missing—at a specified $mask\_ratio$ (e.g., 30%). This process transforms the original sample into $\mathbf{x} = \{\mathbf{x}^{cat}, \mathbf{x}^{num}, \mathbf{x}^{miss}, \mathbf{x}^{masked}\}$, hear $\mathbf{x}^{masked}$ denotes the simulated missing values. We initialize a parameter-shared, learnable mask embedding for both true and simulated missing values. In Sec 3.5, we present a joint learning approach that integrates missing feature reconstruction with downstream task optimization, to improve the applicability of the mask embedding for downstream applications. Furthermore, it is essential to carefully design the masking strategy to mitigate significant shifts in data distribution, which could negatively impact the performance of downstream tasks. We investigate the effective criteria of masked strategy in Sec 4, provide valuable insights for applications across various real-world scenarios characterized by feature missingness.

## 3.3 Feature Embedding Module

Tabular data consists of various data types, predominantly characterized by cat features (including categorical and text features), num features (numerical features), and features with missing values. To improve the understanding of the semantic information associated with features, we employ a pre-trained BERT model to extract embeddings for the feature names. Additionally, to enhance the alignment between the learned mask embedding and the embeddings of both num and cat features, we propose a unified approach for integrating embeddings of feature names and feature values.

Specifically, we treat feature names and cat feature values as text. These are processed sequentially through tokenization, word embedding, and pooling to obtain their embedded representations, as illustrated in the following formulas:

$$\boldsymbol{c}^j = \text{pooling}(\text{emb}(\text{tokenize}(c^j))), j \in \{0, 1, 2, \ldots, a + b + k\}, \tag{1}$$

$$\boldsymbol{x}^{j \in cat} = \text{pooling}(\text{emb}(\text{tokenize}(x^{j \in cat}))). \tag{2}$$

Next, For num features, we employ a separate linear layer for each feature to convert individual values into embeddings, as formula below:

$$\boldsymbol{x}^{j \in num} = \text{Linear}_j(x^{j \in num}). \tag{3}$$

Notably, we initialize a shared learnable mask embedding $\boldsymbol{x}_{miss}$ to represent missing values.

Finally, we concat value embeddings of different data types into $\boldsymbol{X}$, then combine the feature name embeddings $\boldsymbol{C}$ with the value embeddings $\boldsymbol{X}$ in a unified manner, as following:

$$\boldsymbol{E} = \boldsymbol{C} + \text{concat}(\boldsymbol{X}_{cat}, \boldsymbol{X}_{num}, \boldsymbol{X}_{miss}), \tag{4}$$

where $\boldsymbol{E} \in \mathbb{R}^{m \times d}$ represent all feature embeddigns serve as inputs to the transformer model, $m$ and $d$ represent number of features and embedding dimensions respectively.

## 3.4 Design of Tabular Transformer in feature missing scenarios

**Transformer Backbone.** To facilitate comprehensive interactions among high dimensional features, and to learn the latent representations of missing features by leveraging information from other visible features, we employ an encoder-only transformer architecture that utilizes Multi-Head Self-Attention (MHSA) as the feature interaction module. By incorporating feature names into the feature

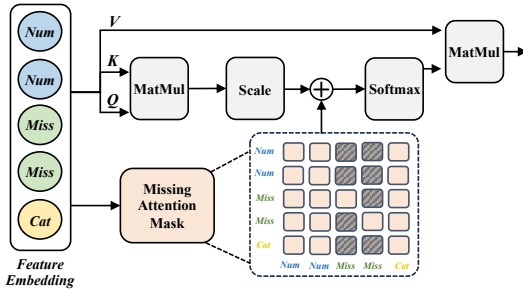

Figure 2: Illustrating the missing-related masked attention mechanism. We establish a masked attention computation for each sample, informed by prior information concerning feature missingness.

embeddings, we eliminate the necessity for positional encoding. Furthermore, we augment the Feed-Forward Network (FFN) structure within the transformer encoder layer by substituting the original ReLU activation function with SwiGLU. The gating mechanisms inherent in SwiGLU enhance the transformer's capacity to evaluate the significance of features automatically.

The feature embeddings $\boldsymbol{E}$ are input into the transformer backbone, yielding the transformed feature representation $\boldsymbol{H}$. Subsequently, $\boldsymbol{H}$ is treated as a token sequence and processed by a mask head composed of linear layers. For num features, the linear layer has dimensions of $d \times 1$, whereas for cat features, the dimensions are of $d \times d$. Additionally, $\boldsymbol{H}$ is processed through pooling operations along the feature dimension to obtain an aggregated representation of each sample, then input into a task head to generate downstream prediction.

**Missing-Related Masked Attention Mechanism.** Specifically, we incorporate prior information about missing features into the computation of attention. This approach can alleviates the impact of missing features on self-attention calculations, thereby preserving the importance of other critical features, which can help the model rely more on trustworthy features when making decisions. We achieve this by masked attention computation, as following formula.

$$\text{Attention}(\boldsymbol{Q}, \boldsymbol{K}, \boldsymbol{V}) = \text{Softmax}(\frac{\boldsymbol{Q}\boldsymbol{K}^T}{\sqrt{d}} + \boldsymbol{M})\boldsymbol{V}, \tag{5}$$

for ease of description, we represent the commonly used query, key, and value in self-attention as $\boldsymbol{Q}$, $\boldsymbol{K}$ and $\boldsymbol{V}$ respectively. Additionally, $\boldsymbol{M}$ denotes the mask derived from the information on missing features, as depicted in Fig. 2, which can be defined as follows:

$$\boldsymbol{M}_{j,k} = \begin{cases} -\infty & \text{if } k \in miss \text{ and } k \neq j \\ 0 & \text{else} \end{cases} \tag{6}$$

### 3.5 JOINTLY APPROACH MISSING FEATURE MODELING AND DOWNSTREAM TASKS

As outlined in Section 3.2, we randomly drop specific features to create simulated missingness, which serves as a supervisory signal for feature reconstruction. Since both true missingness and simulated missingness are represented by mask embedding $\boldsymbol{x}_{miss}$, and the embedding is optimized during joint training, this process facilitates enhanced representations of missing features during downstream tasks. We implement reconstruction loss to optimize the distance between predicted values and original values. For cat and num features, we apply the following two losses:

$$L_{\text{mask}}^{\text{cat}} = \sum_{j \in \mathbb{S}} 1 - \frac{\hat{\boldsymbol{e}}_j \cdot \boldsymbol{e}_j}{||\hat{\boldsymbol{e}}_j||||\boldsymbol{e}_j||}, \tag{7}$$

$$L_{\text{mask}}^{\text{num}} = \sum_{j \in \mathbb{D}} (\hat{x}_j - x_j)^2, \tag{8}$$

Table 1: The details of datasets. We reported the sample size, number of numerical features (#num features), and number of categorical features (#cat features) for 8 datasets. Additionally, we also collected information on the missing condition for each dataset, categorizing the feature missing rates into 3 bins and counting the sample proportion for each bins.

| task type | dataset name | #samples | #num features | #cat features | sample proportion at feature missing rates | | |
|---|---|---|---|---|---|---|---|
| | | | | | (0%, 33%] | (33%, 66%] | (66%, 100%] |
| Classification | Ecom Offers | 160,057 | 113 | 6 | 0.00% | 0.00% | 0.00% |
| | Homesite Insurance | 260,753 | 253 | 46 | 89.93% | 0.00% | 0.00% |
| | HomeCredit Default | 381,664 | 612 | 84 | 57.16% | 36.93% | 5.91% |
| Regression | Sberbank Housing | 28,321 | 365 | 27 | 100.00% | 0.00% | 0.00% |
| | Cooking Time | 319,986 | 186 | 6 | 99.10% | 0.00% | 0.00% |
| | Delivery ETA | 350,516 | 221 | 2 | 91.50% | 3.71% | 0.00% |
| | Maps Routing | 279,945 | 984 | 2 | 97.08% | 2.82% | 0.10% |
| | Weather | 423,795 | 100 | 3 | 0.00% | 2.21% | 0.03% |

where $\mathbb{S}$ and $\mathbb{D}$ represent the cat and num features, respectively, that are selected for masking in a given sample. Moreover, we optimize the objectives of downstream tasks in conjunction with the previously mentioned reconstruction loss, as following:

$$L = L_{\text{task}} + L_{\text{mask}}^{\text{cat}} + L_{\text{mask}}^{\text{num}}, \tag{9}$$

we use Mean Square Error and Cross Entropy loss for regression and classification task respectively.

## 4 EXPERIMENTS

In this section, we validated the effectiveness of our method on the TabReD (Rubachev et al., 2024), a industry-grade tabular benchmark. The experiments demonstrate that the `MaskTab` achieves superior performance on both classification and regression tasks. Furthermore, we conducted ablation experiments on each module to verify the effectiveness of our method.

### 4.1 EXPERIMENTAL DETAILS

**Datasets**. We employed the TabReD benchmark, which exclusively consists of tabular data obtained from real-world industrial applications. This benchmark comprises 8 datasets, including 3 classification tasks and 5 regression tasks. As outlined in Table 1, each dataset is characterized by a substantial number of samples and features. Furthermore, we evaluated the rates of missing features for each dataset and found that most exhibit varying degrees of incompleteness. Additionally, these datasets were divided into training, validation, and testing sets based on temporal criteria. These characteristics are frequently encountered in industrial applications, rendering this benchmark particularly suitable for assessing model performance in such contexts.

**Data Pre-processing**. Follwing FT-transformer (Gorishniy et al., 2021), we categorize the columns of the table into numerical and categorical features. Numerical features are processed through quantile normalize using the Scikit-learn library, with normalization parameters calculated solely on the training set. For regression tasks, we apply the standardization to the labels. Notably, in the experiments conducted with `MaskTab`, we did not fill with any value for missing features.

**Experimental Setup**. We configured the model using default parameters informed by empirical evidence, incorporating 4 transformer blocks, each equipped with 8 attention heads, a feature embedding size of 128, and a dropout rate of 15% within the feed-forward network. The batch size was set at 128, and we employed the AdamW optimizer with a learning rate of $1e-4$. In the vicinity of the default parameters, Hyperparameters were optimized for each dataset through grid search, guided by performance metrics derived from the validation set, notably, the test set was kept untouched during this tuning process. Experiments were conducted on 8 NVIDIA A100 GPUs, and the optimal model checkpoint was selected via early stopping, based on validation metrics. For evaluation, we utilized the receiver operating characteristic area under the curve (ROC AUC) for classification tasks and the root mean square error (RMSE) for regression tasks.

Table 2: Comparing `MaskTab` with current Gradient Boosted Decision Tree (GBDT) methods and deep tabular models using the TabReD benchmark. The entries highlighted in bold indicate the best-performing models.

| Method | Classification (ROC AUC ↑) | | | Regression (RMSE ↓) | | | | | Avg. Rank |
|---|---|---|---|---|---|---|---|---|---|
| | Homesite Insurance | Ecom Offers | HomeCredit Default | Sberbank Housing | Cooking Time | Delivery ETA | Maps Routing | Weather | |
| XGBoost | 0.9601 | 0.5763 | **0.8670** | 0.2419 | 0.4823 | 0.5468 | **0.1616** | 1.4671 | 4.3 |
| LightGBM | 0.9603 | 0.5758 | 0.8664 | 0.2468 | 0.4826 | 0.5468 | 0.1618 | 1.4625 | 5.4 |
| CatBoost | 0.9606 | 0.5596 | 0.8621 | 0.2482 | 0.4823 | **0.5465** | 0.1619 | 1.4688 | 5.6 |
| MLP | 0.9500 | 0.6015 | 0.8545 | 0.2508 | 0.4820 | 0.5504 | 0.1622 | 1.5470 | 6.8 |
| SNN | 0.9492 | 0.5996 | 0.8551 | 0.2858 | 0.4838 | 0.5544 | 0.1651 | 1.5649 | 10.5 |
| DCNv2 | 0.9392 | 0.5955 | 0.8466 | 0.2770 | 0.4842 | 0.5532 | 0.1672 | 1.5782 | 11.4 |
| ResNet | 0.9469 | 0.5998 | 0.8493 | 0.2743 | 0.4825 | 0.5527 | 0.1625 | 1.5021 | 8.0 |
| MLP-PLR | 0.9621 | 0.5957 | 0.8568 | 0.2438 | 0.4812 | 0.5527 | **0.1616** | 1.5177 | 4.4 |
| Trompt | 0.9546 | 0.5792 | 0.8381 | 0.2596 | 0.4834 | 0.5563 | 0.1652 | 1.5722 | 11.0 |
| FT-Transformer | 0.9622 | 0.5775 | 0.8571 | 0.2440 | 0.4820 | 0.5542 | 0.1625 | 1.5104 | 6.4 |
| TabR | 0.9522 | 0.5850 | 0.8484 | 0.2851 | 0.4825 | 0.5541 | 0.1637 | **1.4622** | 8.6 |
| TabNet | 0.9531 | 0.5855 | 0.7701 | 0.2828 | 0.4813 | 0.5567 | 0.1651 | 1.5877 | 10.5 |
| TransTab | 0.9564 | 0.5868 | 0.8498 | - | - | - | - | - | - |
| CM2 | 0.9560 | 0.5890 | 0.8392 | **0.2287** | 0.4838 | 0.5569 | 0.1638 | 1.5339 | 9.0 |
| **MaskTab** (ours) | **0.9635** | **0.6016** | 0.8660 | 0.2337 | **0.4806** | 0.5495 | 0.1618 | 1.4883 | **2.5** |

**Compared Methods**. To adequately validate the effectiveness of our method, we compare it with two major categories of methods: Gradient Boosted Decision Trees (GBDT) and Deep Learning methods. The former includes XGBoost (Chen & Guestrin, 2016), LightGBM (Ke et al., 2017), and CatBoost (Prokhorenkova et al., 2018), while the latter has MLP (Gorishniy et al., 2021), SNN (Klambauer et al., 2017), ResNet (Gorishniy et al., 2021), DCNv2 (Wang et al., 2020), TabNet (Arik & Pfister, 2021) , MLP-PLR (Gorishniy et al., 2022),Trompt (Chen et al., 2023), as well as attention-based method, FT-Transformer (Gorishniy et al., 2021). Further, we include a recent advanced method named TabR (Gorishniy et al., 2024), a retrieval-based model, demonstrated impressive performance. Additionally, we also compared pre-trained method like TransTab (Wang & Sun, 2022) and CM2 (Ye et al., 2024). For all methods, we applied the same training set, validation set, and test set, and calculate the average ranking across 8 dataset.

## 4.2 OVERALL PERFORMANCE

The comparative results of `MaskTab` and other methods are presented in Table 2. By calculating the average ranking across 8 datasets, our method achieves an average rank of 2.5. This performance surpasses XGBoost, the preeminent algorithm among GBDT methods, and significantly exceeds that of other deep tabular models. In comparison to GBDT-based methods, our model demonstrates comparable performance or slight improvements across various datasets. Specifically, in classification tasks involving datasets Homesite Insurance and Ecom Offers, we achieved AUC values of 0.9635 and 0.6016, respectively, outperforming three GBDT methods. Additionally, in the regression task utilizing dataset Sberbank Housing, our model attained an RMSE of 0.2337, surpassing GBDT methods, which generally produce RMSE values exceeding 0.24. Compared to deep learning methods, our approach demonstrates a significant advantage on dataset HomeCredit Default. Specifically, our method achieves an AUC of 0.8660, while other deep tabular models exhibit AUC values below 0.860, with the highest being 0.8571 for FT-Transformer. Notably, this dataset contains 612 numerical features and 84 categorical features, and it experiences considerable feature missingness, with 36.93% of samples have a missing rate ranging from 33% to 66%. The superior performance of our method underscores its effectiveness in contexts characterized by a high dimensionality and considerable missing data, thereby positioning it as a promising option for industrial applications.

## 4.3 ABLATION STUDY

In this section, we conducted ablation studies to systematically evaluate the contribution of each component. We selected two representative datasets, HomeCredit Default and Sberbank Housing,

Table 3: Ablation studies to verify the effectiveness of each component in `MaskTab`, including the joint learning approach, learned mask embedding, and the missing-related masked attention.

| Module | HomeCredit Default (ROC AUC ↑) | Sberbank Housing (RMSE ↓) |
|---|---|---|
| *Training Objective* | | |
| only task training | 0.8634 | 0.2862 |
| masked training → task training | 0.8616 | 0.2366 |
| **masked training + task training** | **0.8660** | **0.2337** |
| *Imputation Strategy* | | |
| zero value | 0.8610 | 0.2405 |
| mode value | 0.8599 | 0.2390 |
| HyperImpute | - | 0.2468 |
| ReMasker | 0.8593 | 0.2469 |
| **mask embedding** | **0.8660** | **0.2337** |
| *Missing-Related Masked Attention* | | |
| w/o | 0.8640 | 0.2371 |
| in first transformer layer | 0.8628 | 0.2364 |
| in all transformer layers | 0.8605 | 0.2527 |
| **in last transformer layer** | **0.8660** | **0.2337** |

both of which exhibit a high rate of missing features. The former corresponds to a classification task, while the latter pertains to a regression task.

**Joint Masked Training and Task Training**. To rigorously validate the advantages of joint training, we devised two supplementary training methodologies for comparison: only task training and a two-stage approach consisting of masked training followed by task training. All three training methods preserving the other improvements to ensure fair evaluation. From the "training objective" section of Table 3, it is evident that the two-stage approach did not enhance the model's performance on HomeCredit Default compared with only task training. This suggests that masked training, by itself, does not effectively promote the development of representations advantageous for downstream tasks. Conversely, joint training produced optimal performance across both datasets.

**Mask Embedding vs. Imputation**. In comparing various imputation methods, we employed joint training while excluding the masked embedding for missing features, retaining all other modules to ensure fairness. The imputation techniques analyzed included mean imputation, mode imputation, and two advanced model-based approaches. As shown in the "Imputation Strategy" section of Table 3, the mask embedding learned by `MaskTab` effectively captured the characteristics of missing values, yielding superior performance across both datasets. In implementing the Hyperimpute on HomeCredit Default, we encountered difficulties due to the excessively large volume of data. Additionally, training the imputation model ReMasker presented challenges in parameter tuning, which resulted in suboptimal performance on downstream tasks. In contrast, our method is more straightforward and facilitates practical usability in real-world applications.

**Missing-Related Masked Attention**. We examine the effectiveness of the missing-related masked attention, as indicated in the final section of Table 3. The results indicate that this module is not suitable for integration at every layer of the Transformer architecture. Instead, its application in the final layer—where critical decision-making occurs—demonstrates optimal performance in downstream tasks. We hypothesize that this module functions to accentuate reliable features, playing a crucial role during the model's decision-making phase, without interfering with the learning of shallow feature interactions in the earlier layers.

**Performance under Different Missing Rates**. To illustrate the significant utility of `MaskTab` in addressing samples with missing features, we partitioned the test set into three subsets of equal size, categorized by low, medium, and high missing rates. We assessed the performance of downstream tasks across these three subsets. As indicated in Table 4, we compared `MaskTab` with a version that omits all improved modules. The results demonstrate that our proposed modules yield a more substantial enhancement in the high missing rate category. Specifically, for HomeCredit Default, the AUC improvement for samples in the high missing rate subset was 0.0031, in contrast to 0.0015 for

Table 4: Performance of `MaskTab` at varying missing rates. The terms low, medium, and high refer to three distinct levels of feature missing, which partition the test set into three equal-sized subsets. The term `MaskTab` * denotes a variant of our method that excludes all enhancements.

| Method | HomeCredit Default (ROC AUC ↑) | | | Sberbank Housing (RMSE ↓) | | |
|---|---|---|---|---|---|---|
| | low | medium | high | low | medium | high |
| XGBoost | 0.8563 | 0.8770 | 0.8656 | 0.2402 | 0.2899 | 0.1892 |
| `MaskTab` * | 0.8561 | 0.8747 | 0.8595 | 0.2322 | 0.2889 | 0.2203 |
| `MaskTab` | 0.8576 | 0.8765 | 0.8626 | 0.2258 | 0.2821 | 0.1826 |
| Relative Improvement | +0.0015 | +0.0018 | **+0.0031** | -0.0064 | -0.0068 | **-0.0377** |

those in the low missing rate group. Similarly, in Sberbank Housing, the RMSE improvements were 0.0377 versus 0.0064, respectively.

**Masking Strategy Analysis**. To examine the relationship between the feature missing rate and the employed masking strategy, we selected two datasets: HomeCredit Default, characterized by a substantially higher missing rate, and Ecom Offers, which exhibits a missing rate of zero. We conducted experiments using a range of mask probs ($[0.25, 0.50, 0.75, 1.0]$) alongside varying mask ratios ($[0.1, 0.3, 0.5, 0.7]$). As shown in Fig. 3, We find that for datasets with a high missing rate, a lower mask prob and mask ratio result in improved performance. Conversely, for datasets with a low missing rate, the model is capable of tolerating a relatively higher level of missing data simulation. We suggest that in the former scenario, the model is required to learn representations of missing data while accommodating a slight shift in the underlying data distribution. In contrast, for the latter, the introduction of additional dropped features act as data augmentation to effectively facilitate the feature learning.

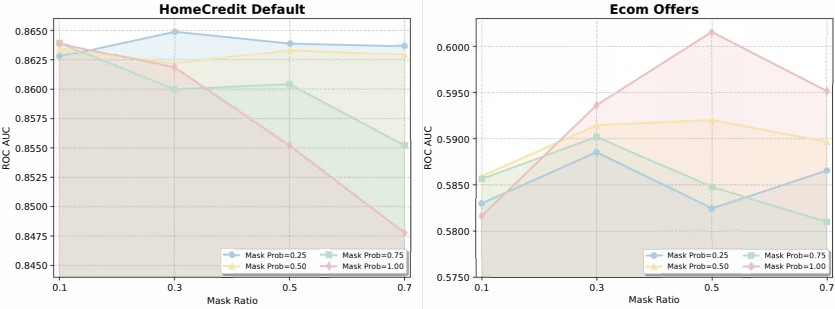

Figure 3: Analysis of masking strategy in `MaskTab`. Different colored curves correspond to various mask prob, while the x-axis represents the mask ratio. The left part presents the analysis results for HomeCredit Default, which exhibits a high rate of missing values. In contrast, the right part illustrates the analysis results for Ecom Offers, characterized by a lower rate of missing values.

## 5 CONCLUSIONS AND FUTURE WORK

In this study, we examine the applicability of deep tabular models in industrial contexts, which are characterized by high dimensionality and significant amounts of missing data. To address the challenge of feature absence, we introduce `MaskTab`, which employs a masked modeling approach for tabular data. Furthermore, our specifically optimized modules exhibit notable effectiveness. Comparative analyses position our method as a promising candidate for industrial applications. A future direction for optimization involves conducting cross-table training using extensive industrial datasets to determine whether larger data scales result in enhanced performance. Additionally, in certain industrial scenarios, some critical features may be entirely absent for specific test samples. We aim to investigate whether training on samples that encompass all features could improve predictive accuracy for those samples with missing key attributes.

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
