# OpenReview forum: "MaskTab: Masked Tabular Data Modeling for Learning with Missing Features"
_ICLR.cc/2025/Conference — ICLR 2025 Conference Withdrawn Submission_

### Official Review · Reviewer_VkKS · 2024-10-29

**Soundness:** 3
**Presentation:** 2
**Contribution:** 2
**Rating:** 6
**Confidence:** 4

**Summary:**

The authors propose an empirical method to impute missing values in tabular data. In particular, they propose to learn an embedding for each missing value -- this embedding is shared across all missing values within a row. To train this embedding, they introduce a reconstruction loss: they simulate missingness and add a loss to predict their values. Finally, the authors also propose a missingness-aware attention block, which masks missing features from others.

**Strengths:**

The paper is interesting because it deals with a frequently overlooked aspect of the tabular data in deep learning, namely missingness. Overall, the approach is simple and could benefit many of the existing Transformer-based tabular models. The results show strong empirical performance against other state-of-the-art methods, including GDBT models such as CatBoost and XGBoost, on an industry-grade benchmark. The authors offer extensive ablation study to justify their architectural choices and offer insights into the model tuning.

**Weaknesses:**

This is purely an experimental paper. As such, a lot of design choices are clearly dictated by performance considerations and are not intuitive or otherwise theoretically justified. I left some questions below, addressing which I think could make the paper stronger.

**Questions:**

1. Is my understanding correct that all of the missing / simulated missing features share the same (learnable) embedding? Are you using any other information about the missing features such as their column names? From my read of the paper, if I have missing features, $\mathbf{x}_{m}$, and masked features, $\mathbf{x}_s$, such that $\[\mathbf{x}_m,\mathbf{x}_s\]\in\mathbb{R}^n$, then your method would embed them as $\[\mathbf{x}_m,\mathbf{x}_s\] \to \[\mathbf{e}^T, \mathbf{e}^T, ..., \mathbf{e}^{T}\]\in\mathbb{R}^{d\times n}$, where $\mathbf{e}$ is your learnable embedding. It would help a lot if more details regarding this were included in Section 3.2 and 3.3.
2. I do not understand how your `num_mask_head` and `cat_mask_head` work from reading Section 3.5 and Figure 1. Since the simulated dropped features could be different between rows within a single batch, how do these heads predict the missing values? I am supposing these heads are reconstructing the whole row, but the loss is only computed for the masked missing values. It would be nice if further clarification is provided in Section 3.5 regarding this.
3. You introduce the missing-related attention mechanism, but empirically it only helps when added in the last layer. Is there any intuition here? Supposedly, if no masking has been used in the previous layers, the information should have been already shared between all the features. So, in a sense, it should not matter if you mask some other positions or mask more tokens.
4. In Section 4.1, how are missing values imputed for the other methods? Some of the models can handle missingness out-of-the-box; however, I suppose for all deep learning methods, some pre-processing had to be applied.
5. In Section 4.3 "Performance under Different Missing Rates", what do you mean by MaskTab without enhancements. If you don't use the missing feature embeddings, how are those missing features handled then?
6. The related works could use some improvement. For example, the recently released, LAMDA-TALENT [1] benchmark includes many more methods. In particular, related to your method, unsupervised masking via contrastive learning for robustifying DNN has been previously explored in SwitchTab [2].

[1] https://github.com/qile2000/LAMDA-TALENT

[2] Wu, Jing, et al. "Switchtab: Switched autoencoders are effective tabular learners." Proceedings of the AAAI Conference on Artificial Intelligence. Vol. 38. No. 14. 2024.

---

### Official Review · Reviewer_bUvz · 2024-10-30

**Soundness:** 2
**Presentation:** 2
**Contribution:** 2
**Rating:** 3
**Confidence:** 4

**Summary:**

The authors propose using a new trend of masked modeling in a tabular setting and compare the results with several simple baselines, such as XGBoost, etc. However, it is not entirely clear what the main challenges of tabular data for such training are that the authors are trying to address. I expected to see some sort of novelty, at least in terms of the masking strategy.

**Strengths:**

The foundation model for tabular data is important.

**Weaknesses:**

- The novelty is somewhat limited, as there are not many specific improvements compared to existing methods in the new tabular setting.
- The experiments are also somewhat limited. For example, baselines such as SSL methods for tabular data, including SubTab, VIME, SCARF, PTaRL, etc are not included.
- The experiments lack standard deviation for the reported values.
- How does the performance compare to foundation models or LLMs?
- Overall, I believe the paper's aims, claims, and experiments do not seem competitive with the current ICLR standards.

**Questions:**

- Please consider adding more baselines, especially those that can handle missing data, including SubTab, etc.
- Clearly outline the main challenges that make this setting different from other data modalities and how you addressed those challenges. Missing data also exists in language and biological data.
- The implementation details are not clear.

---

### Official Review · Reviewer_3iL8 · 2024-11-01

**Soundness:** 1
**Presentation:** 3
**Contribution:** 1
**Rating:** 3
**Confidence:** 3

**Summary:**

This paper proposes a new approach to deal with missing values when the final goal is creating a supervised model. Their approach is based on masking inputs in the training set when training their classifiers.

**Strengths:**

1) Masking is an interesting approach to modeling missing values;
2) The paper is well written.

**Weaknesses:**

Even though the authors approach this problem from an interesting angle (i.e., masking), it suffers from major weaknesses:
1) Their method lacks statistical intuition and/or theory. For example, it is not clear when/why their approach should work.
2) The authors make no distinction on the types of missing patterns they are tackling: is it MAR, MCAR, etc?
3) The empirical results do not seem promising when compared to baselines.

In summary, I do not see an expressive contribution either in terms of theory or empirical results.

**Questions:**

Assuming training and testing have the same missing patterns, why is masking a sensible approach? In my view, masking can be only useful in cases where there is a distribution shift when you are trying to emulate test missing patterns. The authors comment on distribution shifts but I do not think this is explored in depth as it should be (perhaps using a more formal statistical framework).

---

### Official Review · Reviewer_Ygbd · 2024-11-03

**Soundness:** 2
**Presentation:** 2
**Contribution:** 1
**Rating:** 1
**Confidence:** 4

**Summary:**

The paper presents **MaskTab**, a framework for handling missing features in tabular data by incorporating simulated missing data directly into downstream tasks. The authors claim MaskTab outperforms other DNN-based methods on the TabReD benchmark by employing masked attention to prioritise trustworthy features.

**Strengths:**

- Tackles the prevalent problem of missing features in tabular data.
- Shows slight improvement on TabReD benchmark datasets, though the results are somewhat questionable.

**Weaknesses:**

I find the novelty of the proposed method questionable. Masking features during both pre-training and training for downstream tasks is already common practice, often implemented by introducing various forms of noise (e.g., SubTab [1]) or explicitly masking to improve representation learning (e.g., VIME [2]). Additional concerns include:

- **Lack of Confidence Intervals**: Confidence intervals (e.g., error bars) are not reported, which exaggerates the model's performance claims. For instance, in Table-2, MLP and MaskTab appear to perform similarly for the Ecom Offers dataset, and TabNet’s performance is comparable to MaskTab for Cooking Time dataset when considering a 95% confidence interval (though this isn't shown).
- **Incomplete Literature Review**: The paper provides a limited review of tabular data methods and should reference studies that mask or corrupt input features, such as SubTab [1] and VIME [2].
- **Limited Value of "Missing-Related Masked Attention"**: The ablation study shows minimal benefit from the proposed masked attention, likely because attention models inherently prioritize informative features. Explicitly adjusting attention weights based on missing features may not be necessary.
- **Baseline Comparison**: A baseline using existing models with imputed data should be included to clarify the performance contrast and demonstrate any true advantage of the proposed method.

**References:**

[1] SubTab: Subsetting Features of Tabular Data for Self-Supervised Representation Learning \
[2] Vime: Extending the success of self-and semi-supervised learning to tabular domain.

**Questions:**

No questions.

---

### Note · Authors · 2024-12-09

I have read and agree with the venue's withdrawal policy on behalf of myself and my co-authors.